

# PFI-3 induces vasorelaxation with potency to reduce extracellular calcium influx in rat mesenteric artery

Jing Li[1], Xue-Qi Liang[2], Yun-Feng Cui[1], Yu-Yang Fu[1], Zi-Yue Ma[1], Ying-Tao Cui[1], Xian-Hui Dong[1], Hai-Jun Huang[1], Ting-Ting Tong[1], Ya-Mei Zhu[1], Ya-Dong Xue[1], Yong-Zhen Wang[1], Tao Ban[1] and Rong Huo[1]

[1] Department of Pharmacology (The Key Laboratory of Cardiovascular Research, Ministry of Education) at College of Pharmacy, Harbin Medical University, Harbin, Heilongjiang Province, China
[2] Department of Pharmacy, Second Affiliated Hospital of Qiqihar Medical College, Qiqihar, Heilongjiang Province, China

## ABSTRACT

**Background**. PFI-3 is a small-molecule inhibitor that targets the bromodomains (BRDs) of Brahma-related gene 1 (BRG1). This monomeric compound, which has high selectivity and potent cellular effects, has recently been developed. Although PFI-3 has been reported as a potential therapeutic agent targeting thrombomodulin, its role in the regulation of vascular function remains unknown. Therefore, we aimed to investigate the impact of PFI-3 on arterial vessel tone.

**Methods**. A microvascular tension measurement device (DMT) was utilized to identify alterations in vascular tension within the mesenteric artery. To detect variations in cytosolic $[Ca^{2+}]_i$, a Fluo-3/AM fluorescent probe and fluorescence microscope were employed. Additionally, whole-cell patch clamp techniques were utilized to evaluate the activity of L-type voltage-dependent calcium channels (VDCCs) in cultured arterial smooth muscle cells (A10 cells).

**Results**. PFI-3 exerted a dose-dependent relaxation effect on rat mesenteric arteries with both intact and denuded endothelium after phenylephrine (PE)- and high-$K^+$-induced constriction. PFI-3-induced vasorelaxation was not affected by the presence of L-NAME/ODQ or $K^+$ channel blockers (Gli/TEA). PFI-3 abolished $Ca^{2+}$-induced contraction on endothelium-denuded mesenteric arteries preincubated by PE in $Ca^{2+}$-free solution. Incubation with TG had no impact on PFI-3-induced vasorelaxation pre-contracted by PE. PFI-3 reduced $Ca^{2+}$-induced contraction on endothelium-denuded mesenteric arteries pre-incubated by KCl (60 mM) in $Ca^{2+}$-free solution. PFI-3 declined extracellular calcium influx in A10 cells detected by Fluo-3/AM fluorescent probe and fluorescence microscope. Furthermore, we observed that PFI-3 decreased the current densities of L-type VDCC by whole-cell patch clamp techniques.

**Conclusions**. PFI-3 blunted PE and high $K^+$-induced vasoconstriction independent of endothelium on rat mesenteric artery. The vasodilatory effect of PFI-3 may be attributed to its inhibition of VDCCs and receptor-operated calcium channels (ROCCs) on vascular smooth muscle cells (VSMCs).

Corresponding authors
Tao Ban, bantao2000@163.com
Rong Huo, huohuoz010603@163.com

## INTRODUCTION

PFI-3 is a highly potent and specific cell-permeable probe with negligible cytotoxicity (*Fedorov et al., 2015*; *Lee et al., 2021*). This inhibitor (*Vangamudi et al., 2015*) selectively targets the bromodomains (BRDs) in the C-terminus of chromatin recombinant Brahma-related gene 1 (BRG1), affecting the expression of downstream genes regulated by BRG1 (*Filippakopoulos et al., 2012*). Thus far, PFI-3 has been confirmed to have a crucial effect in both cancer and vascular disease. PFI-3 significantly enhances the antitumor effect of TMZ on intracranial glioblastoma multiforme (GBM) animal models, causing a noticeable increase in the survival of animals with GBM tumors (*Yang et al., 2021*). Another recent study reported that PFI-3 reduced the growth of t (4;14) xenograft tumors (*Chong et al., 2021*). *Zhang et al. (2020)* demonstrated that PFI-3 inhibited the inflammatory response by alleviating tumor necrosis factor-$\alpha$-induced IL-6 and CCL2 expression, decreasing c-Fos expression, and blocking c-Fos translocation into nuclei. Moreover, *Wu et al. (2022)* found that PFI-3 treatment alleviated the downregulation of thrombomodulin in endothelial cells and reduced the incidence of deep vein thrombosis induced by surgical procedures in mice.

It is currently unclear whether PFI-3 affects vessel tone *via* endothelial cells or vascular smooth muscle cells (VSMCs). Our objective in this study was to explore the effects of PFI-3 on vascular function of rat mesenteric arteries and elucidate the potential mechanisms.

## MATERIALS AND METHODS

### Agents

PFI-3 (E)-1-(2-hydroxyphenyl)-3-((1R,4R)-2-pyridin-2-yl-2,5-diazabicyclo[2.2.1] heptan-5-yl) prop-2-en-1-one, ODQ 1H-[1,2,4]oxadiazolo[4,3-a]quinoxalin-1-one, thapsigargin (TG) and phenylephrine (PE) were purchased from Sigma-Aldrich (Saint Louis, MO, USA). Acetylcholine chloride (Ach) was purchased from Harvest Pharmaceutical Co. Ltd. (Shanghai, China). L-NAME, N($\omega$)-nitro-L-arginine methyl ester, glibenclamide (Gli), and tetraethylammonium (TEA) were purchased from MedChemExpress LLC (Shanghai, China). Fluo-3/AM was purchased from Life Technologies (Invitrogen, Waltham, MA, USA). Ach, PE, physiological salt solution (PSS), and high-K$^+$ salt solution (KPSS) were dissolved in double distilled water, and PFI-3, ODQ, L-NAME, TG, Gli and TEA were dissolved in DMSO (Tianjin Fuyu Fine Chemical Co., Ltd, Wuqing District, China). Arterial smooth muscle cells (A10) were purchased from the American Type Culture Collection (ATCC). KPSS, was composed of (in mM): 74.4 NaCl, 60 KCl, 1.17 MgSO$_4$ · 7H$_2$O, 1.18 KH$_2$PO$_4$, 14.9 NaHCO$_3$, 1.6 CaCl$_2$, 5.5 D-glucose, and 0.026 EDTA. The PSS solution was composed of (in mM): 130 NaCl, 4.7 KCl, 1.17 MgSO$_4$ ·7H$_2$O, 1.18 KH$_2$PO$_4$, 14.9 NaHCO$_3$, 1.6 CaCl$_2$, and 5.5 D-glucose.

### Animals and mesenteric artery tension measurement

Adult male Sprague-Dawley (SD) rats (12 weeks old, male, body weight 320–350 g) were purchased from the Experimental Animal Center of Harbin Medical University (Grade II). The adult male SD rats were maintained on a standard light/dark cycle (12 h/12 h)

at 30–70% humidity and a constant temperature ($23 \pm 3\,°C$). Standard chow and water were freely provided to the SD rats. All animal experiments were performed in accordance with NIH guidelines (Guide for the Care and Use of Laboratory Animals) and approved by the Institutional Animal Care and Use Committee of Harbin Medical University (IRB3102619). The adult male SD rats were anesthetized with sodium pentobarbital (40 mg/kg, ip). The animals were euthanized *via* cervical dislocation following anesthesia, and then the intact mesenteric artery was then quickly isolated and transferred to a cold PSS (at $4\,°C$) preoxygenated with 95% $O_2$ and 5% $CO_2$. The adipose tissue and residual blood on the mesenteric artery were promptly removed, and a 2 mm long section of the vessel ring was excised. To construct an endothelium-denuded vascular ring model, the mechanical method was employed; specifically, rat whiskers with a minimum diameter of 80 µm were slowly rubbed on the inner surface of the vascular ring to remove the endothelial cells. The absence of vasorelaxation in response to 1 µM Ach indicated successful endothelial cell disruption. The mesenteric artery ring was fixed with two metal wires threaded through it in a bath containing a precooled PSS solution continuously supplied with 95% $O_2$ and 5% $CO_2$ using a microvascular tension measurement device (DMT620, Danish Myo Technology, Aarhus, Denmark). The mesenteric artery rings were allowed to equilibrate for 60 min before undergoing a wake-up procedure involving reactivation of the mechanical, functional, and signaling functions of the vessel *via* stimuli with PE and KPSS. The isometric systolic tension of the mesenteric artery was recorded using a DMT620 microvascular tension measurement device.

## MTT assays

After the cells adhered to 96-well plates, they were treated with varying concentrations of PFI-3. Subsequently, 200 µL of MTT solution was added to each well and the cells were incubated for 20 min at $37\,°C$. After incubation, the supernatant was removed, and 150 µL of DMSO was added to dissolve the formazan crystals formed by the viable cells. The optical density was measured at 490 nm using an absorbance spectrophotometer.

## Measurement of the cytosolic $[Ca^{2+}]_i$ in VSMCs

A10 cells were cultured and washed three times with serum-free high glucose DMEM. Then, 5 µM fluo-3/AM was added and the cells were incubated at $37\,°C$ in darkness for 30 min. The culture medium was discarded, and the cells were washed three times with warm $Ca^{2+}$-free KPSS (at $37\,°C$). Subsequently, 1 ml of warm calcium-free KPSS (at $37\,°C$) was added, and the fluorescence intensity of fluo-3 in the cells was measured.

## Whole-cell patch clamp experiments

All whole-cell patch-clamp experiments were performed using an Axopatch 700B amplifier. Whole-cell currents were sampled at 1 kHz and filtered at 2 kHz. The $I_{Ca,L}$ in A10 cells was recorded in extracellular solution containing (in mM) 120 TEA, 10 HEPES, 1 $MgCl_2 \bullet 6H_2O$, 10 CsCl, 10 glucose, and 1.8 $CaCl_2$ (pH adjusted to 7.30 with CsOH). The pipette solution contained (in mM) 120 CsCl, 1 $MgCl_2 \bullet 6H_2O$, 40 CsOH, 5 Mg-ATP, 11 EGTA, and 10 HEPES (pH adjusted to 7.30 with CsOH). The cells were maintained at a potential of $-50$ mV.
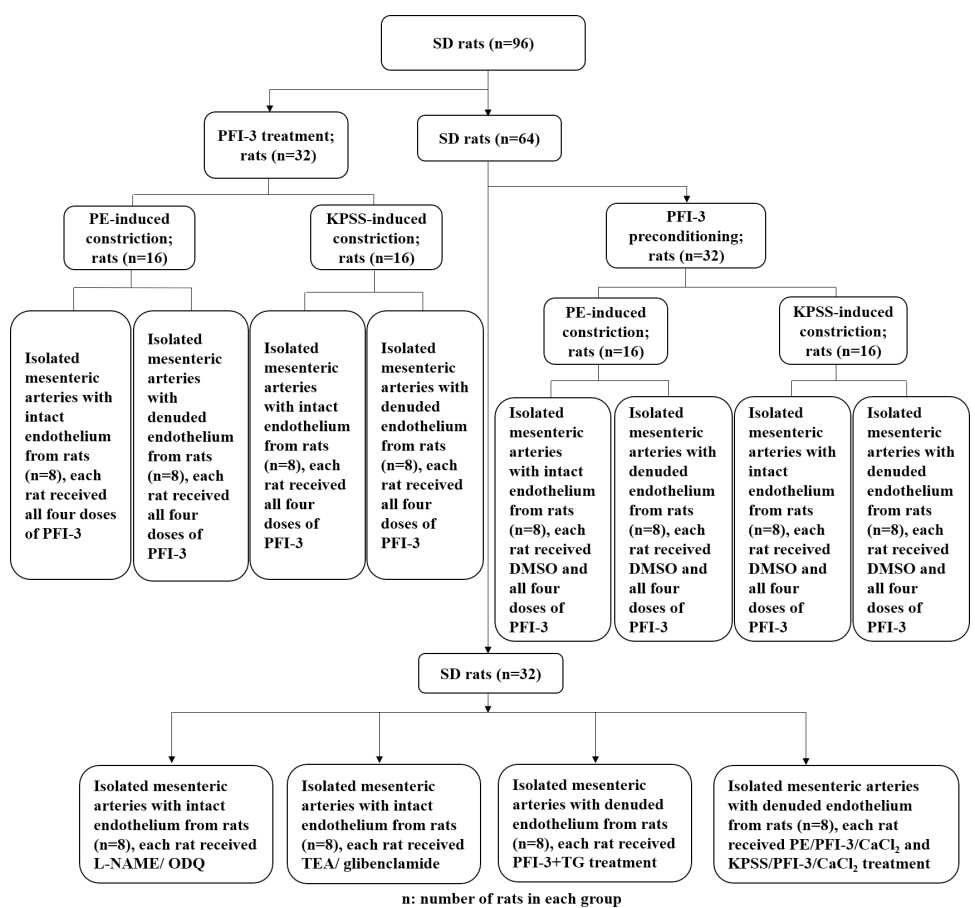

**Figure 1** **Flow chart of the animal experimental procedure.** The procedure of the rat in vitro experiments involving rat tissues. SD, Sprague-Dawley.

## Statistical analysis

All data are presented as mean ± standard deviation (SD). Data were analyzed using the GraphPad Prism (v8.0; GraphPad Software, USA) software.

Two-tailed Student's paired and unpaired $t$ tests were used to analyze the differences between two variables. Differences among groups were evaluated using one-way ANOVA followed by Tukey's test or Dunnett's test for *post-hoc* comparison when appropriate. $P$-values <0.05 were considered statistically significant.

## RESULTS

### PFI-3 did not exhibit cytotoxic effects on A10 cells

The animal experimental design is presented in Fig. 1. To investigate the cytotoxic effect of PFI-3 on A10 cells, the MTT assay was utilized to examine cell viability (Fig. 2). Consistent with previous studies (*Gerstenberger et al., 2016*), PFI-3 did not substantially affect cell viability at doses up to 5 µM. The maximum dose of PFI-3 used in this study was 4 µM.

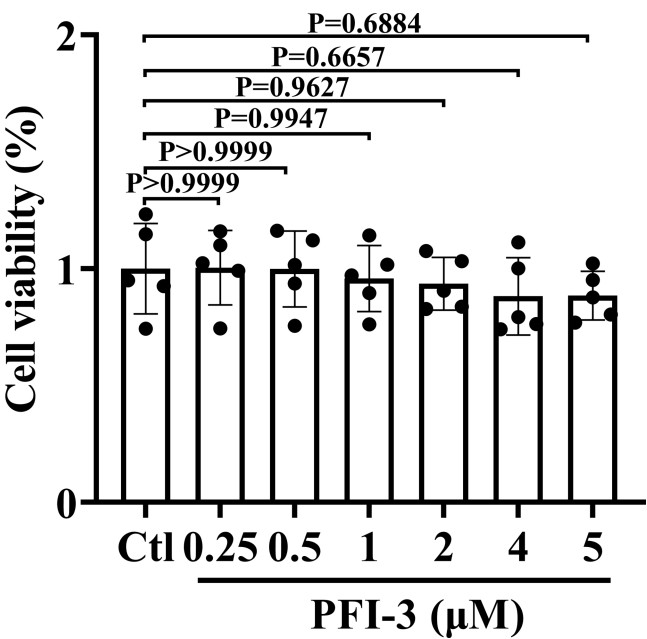

**Figure 2** The cell viabilities of PFI-3 detected by the MTT assay in A10 cells. $P > 0.999$, $P > 0.999$, $P = 0.9944$, $P = 0.9627$, $P = 0.6657$, $P = 0.6884$ vs. the control group; ($n = 5$). Data are represented as the mean $\pm$ SD. One-way ANOVA followed by Tukey's test for *post-hoc* comparisons was used. $n$, number of independent cells.

## PFI-3 dose-dependently induced vasorelaxation of rat mesenteric artery after PE and high $K^+$ -evoked constriction

As depicted in Fig. 3A, the mesenteric arteries were pre-constricted by PE (5 μM), and after the vasoconstriction reached the maximum level and stabilized, the endothelium-dependent vasodilator Ach (1 μM) was administered. The occurrence of Ach-induced relaxation confirmed that the endothelium of the mesenteric arteries was intact. Compared to the DMSO treatment, PFI-3 resulted in concentration-dependent relaxation in rat mesenteric arteries with both intact endothelium (Fig. 3B) and denuded endothelium (Fig. 3D).

Similar to above, high $K^+$ could also induce vasoconstriction of rat mesenteric artery, PFI-3 treatment relaxed the vasoconstriction, regardless of the intact or denuded endothelium (Fig. 4).

## PFI-3 preincubation inhibited PE and high $K^+$-evoked contraction in rat mesenteric artery

The study further investigated the preventive effects of PFI-3 against PE- and high $K^+$-induced vasoconstriction. As shown in Figs. 5–6, pretreatment with PFI-3 (4 μM) for 20 min significantly inhibited PE- and KPSS-induced vasoconstriction in endothelium-intact and -denuded rat mesenteric arteries.

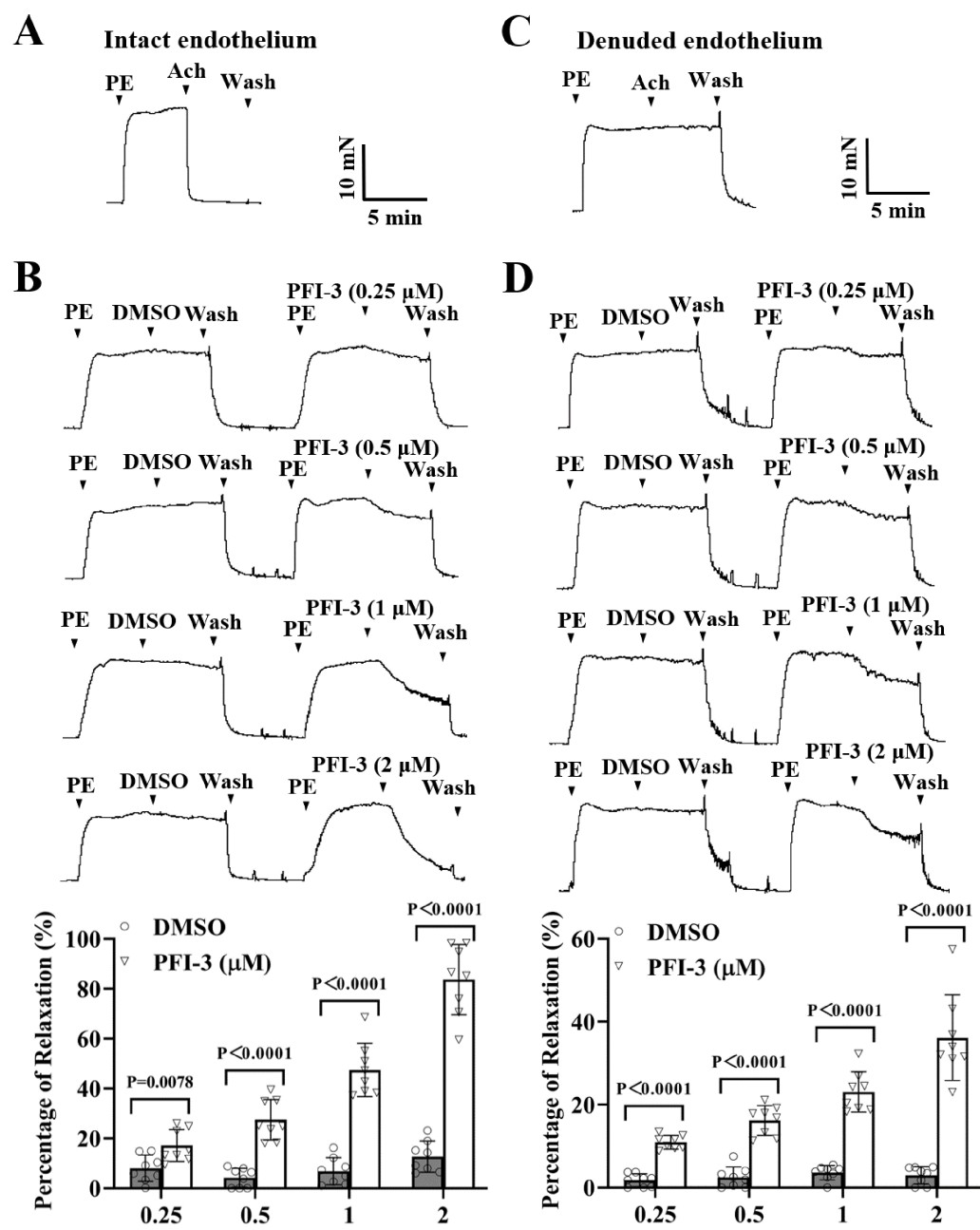

**Figure 3** **PFI-3 induced relaxation of rat mesenteric arteries after PE-induced constriction.** (A–B) The occurrence of Ach (1 μM)-induced vasodilation confirmed that the endothelia of the rat mesenteric arteries was intact. The original traces and summary data demonstrated that unlike DMSO, PFI-3 induced dilation of rat mesenteric arteries with intact endothelium after PE (5 μM)- induced constriction. $P =$ 0.0078, $P < 0.0001$ vs. the DMSO group ($n = 8$). (C) Endothelial denudation of rat mesenteric arteries was confirmed by the absence of Ach (1 μM)-induced vasodilation. (D) The original recording and summary data showed that PFI-3, but not DMSO, induced dilation of mesenteric arteries with denuded endothelium following PE-induced constriction. $P < 0.0001$ vs. the DMSO group ($n = 8$). The error bars represent the SD, and two-tailed unpaired $t$ tests were performed to evaluate the significance of differences. The number of rats or arteries isolated from different rats is indicated as "$n$".

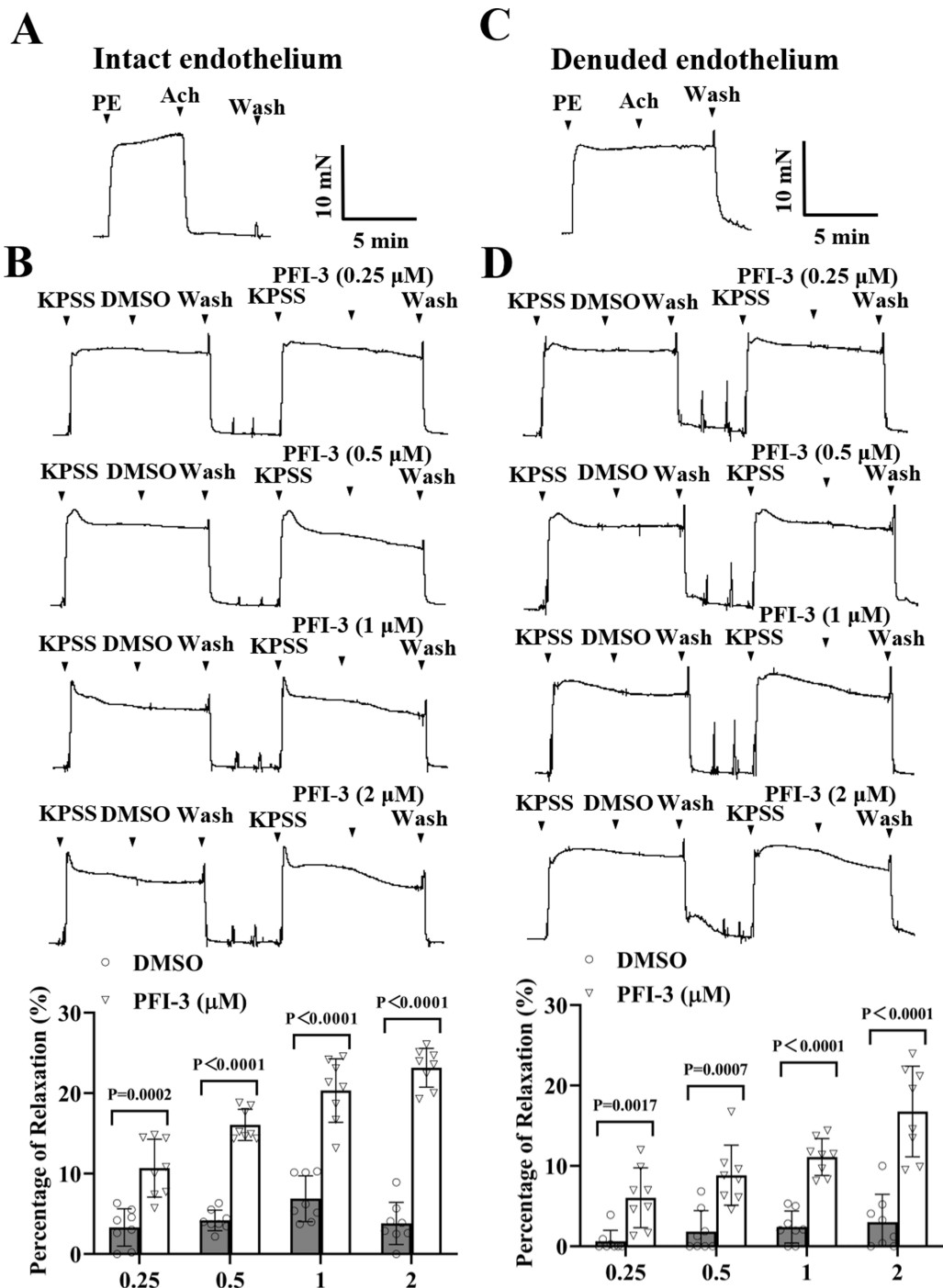

**Figure 4 PFI-3 induced vasodilation of rat mesenteric arteries following KPSS-induced constriction.**
(A) The occurrence of Ach (1 μM)-induced vasodilation confirmed that the endothelia of the rat mesenteric arteries was intact. (B) The original traces and summary data demonstrated that unlike DMSO, PFI-3 induced dilation of rat mesenteric arteries with intact endothelium following KPSS-induced constriction. $P = 0.0002, P < 0.0001$ vs. the DMSO group ($n = 8$). (C) Endothelial denudation of rat mesenteric arteries was confirmed by the absence of Ach (1 μM)-induced vasodilation. (D) The original recordings and summary data showed that PFI-3, but not DMSO, induced dilation of mesenteric arteries with denuded endothelium following KPSS- induced constriction. $P = 0.0017, P = 0.0007, P < 0.0001$ vs. the DMSO group ($n = 8$). The error bars are the SDs. Two-tailed unpaired $t$ tests were performed to evaluate the significance of differences. $n$, the number of rats or arteries isolated from different rats.

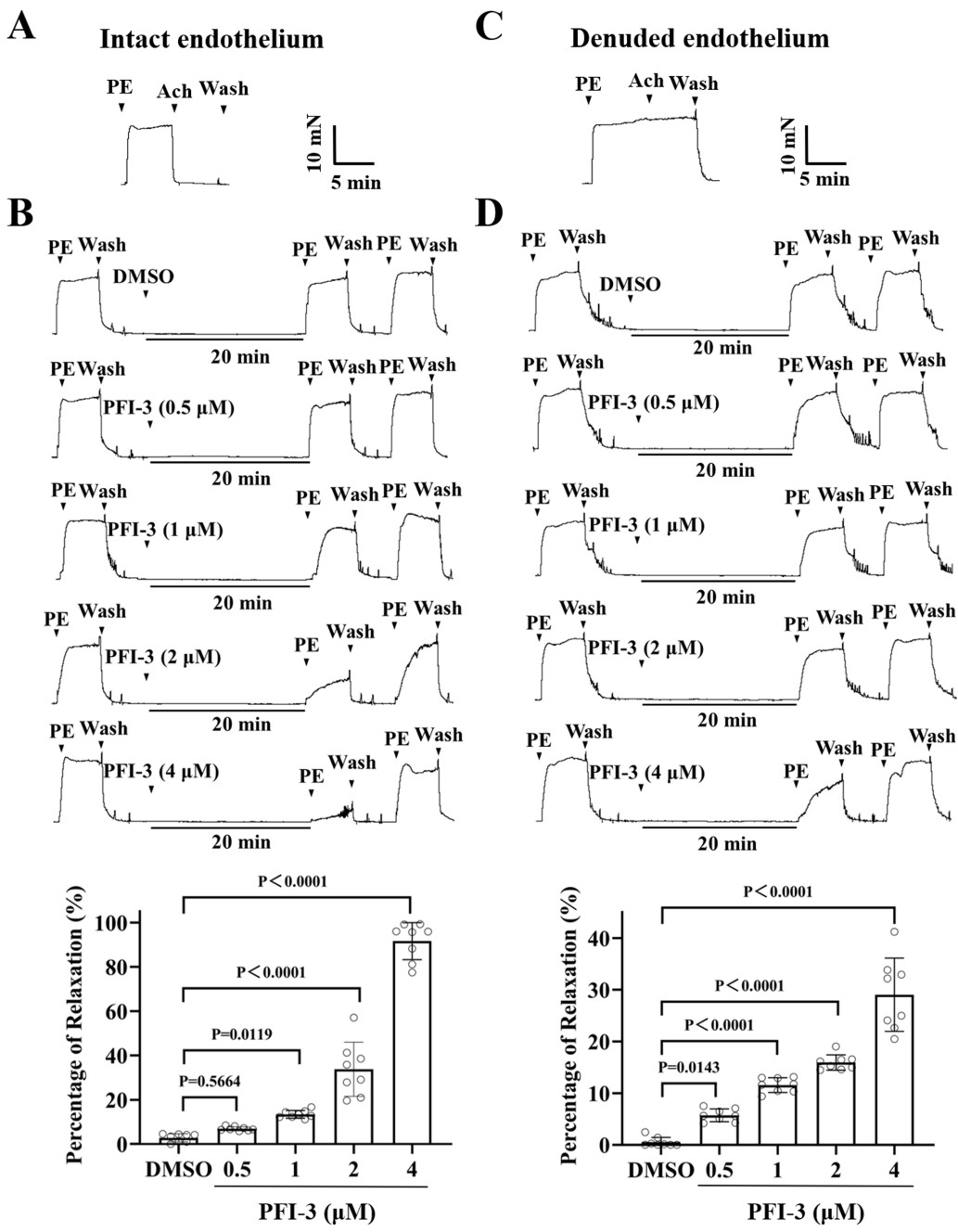

**Figure 5  Pretreatment with PFI-3 prevented PE-induced rat mesenteric artery constriction.** A) The occurrence of Ach (1 μM)-induced vasodilation confirmed that the endothelia of the rat mesenteric arteries was intact. (B) Original traces and summary data showed that unlike DMSO treatment, pretreatment with PFI-3 (4 μM) for 20 min induced dilation of rat mesenteric arteries with intact endothelium following PE (5 μM)-induced constriction. $P = 0.5644, P = 0.0119, P < 0.0001$ vs. the DMSO group ($n = 8$). (C) Endothelial denudation of rat mesenteric arteries was confirmed by the absence of Ach (1 μM)-induced vasodilation. (D) Pretreatment with PFI-3 (4 μM), but not DMSO, for 20 min induced dilation of rat mesenteric arteries with denuded endothelium following PE-induced constriction. $P = 0.0143, P < 0.0001$ vs. the DMSO group ($n = 8$). The error bars are the SDs. One-way ANOVA followed by Dunnett's posttest vs. the DMSO group. $n$, the number of rats or arteries isolated from different rats.

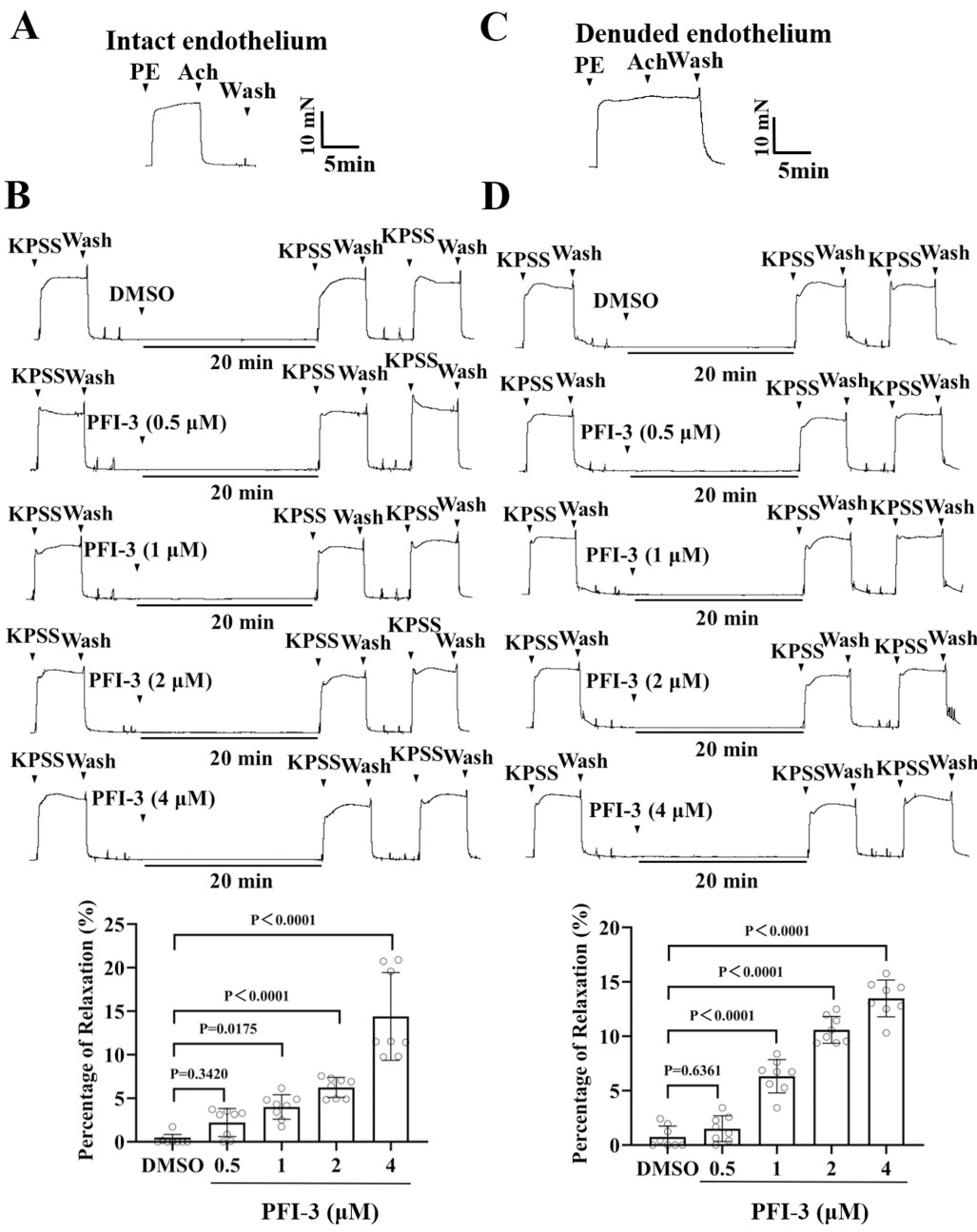

**Figure 6 Pretreatment with PFI-3 prevented KPSS-induced rat mesenteric artery constriction.** (A) The occurrence of Ach (1 µM)-induced vasodilation confirmed that the endothelia of rat mesenteric arteries was intact. (B) Original traces and summary data showed that unlike DMSO treatment, pretreatment with PFI-3 (4 µM) for 20 min induced dilation of the rat mesenteric arteries with intact endothelium following KPSS-induced constriction. $P = 0.3420$, $P = 0.0175$, $P < 0.0001$ vs. the DMSO group ($n = 8$). (C) Endothelial denudation of rat mesenteric arteries was confirmed by the absence of Ach (1 µM)-induced vasodilation. (D) Pretreatment with PFI-3 (4 µM), but not DMSO, for 20 min induced dilation of rat mesenteric arteries with denuded endothelium following KPSS-induced constriction. $P = 0.6361$, $P < 0.0001$ vs. the DMSO group ($n = 8$). The error bars are the SDs. One-way ANOVA followed by Dunnett's posttest vs. the DMSO group. $n$, the number of rats or arteries isolated from different rats.

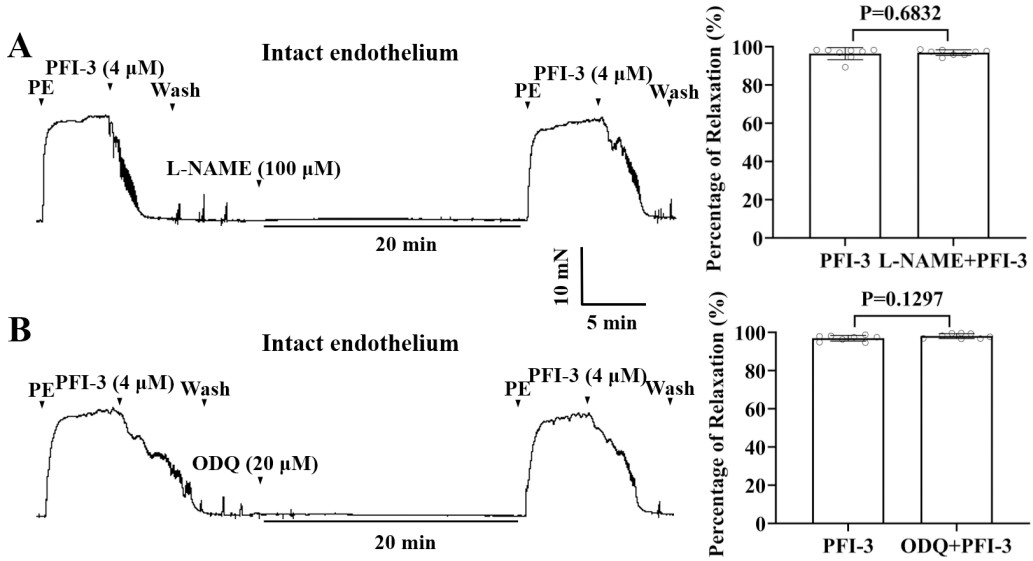

**Figure 7** **PFI-3 induced vasorelaxation independent of the NO/sGC/cGMP pathway.** Effects of L-NAME (100 μM) on PFI-3-induced vasorelaxation of rat mesenteric arteries with intact endothelium. $P = 0.6832$ vs. the PFI-3 group ($n = 8$). (B) Effects of ODQ (20 μM) on PFI-3-induced relaxation of rat mesenteric arteries with intact endothelium. $P = 0.1297$ vs. the PFI-3 group ($n = 8$). The error bars are the SDs. Two-tailed paired $t$ tests were performed to evaluate the significance of differences. $n$, number of rats or arteries isolated from different rats.

### PFI-3 induced vasorelaxation without involving the nitric oxide (NO)/ soluble guanylate cyclase (sGC)/cyclic guanosine monophosphate (cGMP) pathway in rat mesenteric arteries

To assess the influences of the nitric oxide synthase inhibitor, L-NAME, and the guanylate cyclase inhibitor ODQ on PFI-3-induced arterial vasodilation, mesenteric arteries were exposed to L-NAME (100 μM) or ODQ (20 μM) for 20 min. As depicted in Figs. 7A and 7B, neither L-NAME nor ODQ showed any effect on PFI-3-induced vasorelaxation.

### PFI-3-induced vasorelaxation was irrelevant to K⁺ channel blockers in rat mesenteric arteries

Incubation with K⁺ channel blockers, such as Gli (20 μM), or TEA (1 mM), did not alter PFI-3-induced relaxation of endothelium-intact mesenteric arterial rings pre-contracted by PE (5 μM) (Fig. 8).

### PFI-3 affected extracellular Ca²⁺-induced vasoconstriction

High K⁺ induced contractile response of mesenteric arterial rings with denuded endothelium, due to the depolarization of VSMCs and extracellular Ca²⁺ influx through voltage-dependent calcium channels (VDCCs) (*Jones, Jones & Channer, 2003*). In high-K⁺ PSS solution with Ca²⁺-free, addition of $CaCl_2$ (0.3–3 mM) led to a gradual increase of mesenteric arterial ring tension. Pretreatment with PFI-3 reduced the amplitude of contraction induced by extracellular $CaCl_2$ (Fig. 9A). During PE-induced contraction, intracellular calcium stores are mobilized firstly, and this is followed by Ca²⁺ influx

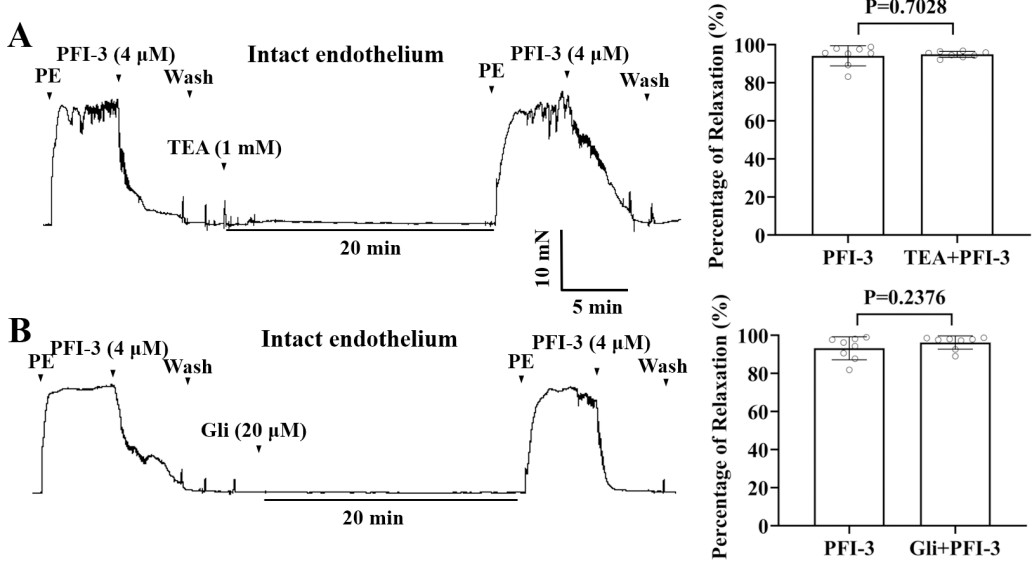

**Figure 8** **The role of the large conductance BKCa channels and $K_{ATP}$ channels in PFI-3-induced vasodilation.** (A) Original traces and summary data showed that preincubation with TEA (1 mM) had no effect on PFI-3-induced vasorelaxation of rat mesenteric arteries with intact endothelium following PE (5 μM)-induced constriction. $P = 0.7028$ vs. the PFI-3 group ($n = 8$). (B) Original traces and summary data showed that preincubation with Gli (20 μM) had no effect on PFI-3-induced vasorelaxation of rat mesenteric arteries with intact endothelium following PE (5 μM)-induced constriction. $P = 0.2376$ vs. the PFI-3 group ($n = 8$). The error bars are the SDs. Two-tailed paired $t$ tests were performed to evaluate the significance of differences. $n$, number of rats or arteries isolated from different rats.

through receptor-operated calcium channels (ROCCs) (*Ehrlich, 1988*; *Arruda-Barbosa et al., 2021*). Mesenteric arterial rings with denuded endothelium were treated with PE (5 μM) without $Ca^{2+}$, and the vessel tone gradually increased with the addition of $CaCl_2$ (0.3–3 mM). However, the increased vessel tone was abolished after PFI-3 pretreatment (Fig. 9B).

## PFI-3 induced vasorelaxation without involving intracellular $Ca^{2+}$ release

To investigate the involvement of $Ca^{2+}$ release from sarcoplasmic reticulum (SR) in PFI-3-induced vasorelaxation, the effect of TG on rat mesenteric artery rings was assessed. Whereas, TG did not show a significant effect on PFI-3-induced vasorelaxation (Fig. 10).

## PFI-3 affected the cytosolic $[Ca^{2+}]_i$ in A10 cells

High $K^+$ induced depolarized of the membrane potential of VSMCs, leading to the influx of extracellular $Ca^{2+}$ through VDCCs (*Betrie et al., 2021*). To further elucidate the effect of PFI-3 on high-$K^+$-induced $Ca^{2+}$ influx, the cytosolic $[Ca^{2+}]_i$ was measured in A10 cells using a Fluo-3/AM fluorescent probe. As illustrated in Fig. 11, in $Ca^{2+}$-free KPSS, PFI-3 did not affect the cytosolic $[Ca^{2+}]_i$, but after treatment of $CaCl_2$ (2 mM) PFI-3 decreased cytosolic $[Ca^{2+}]_i$.

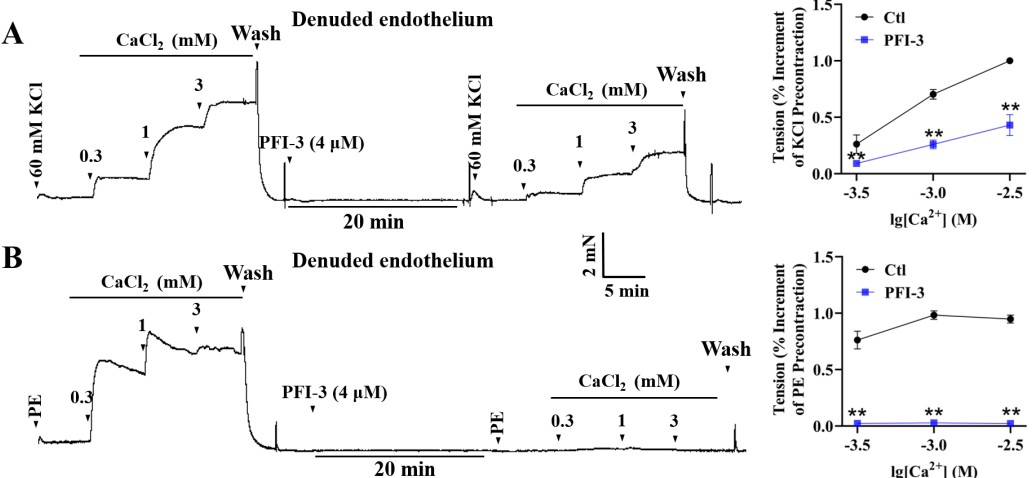

**Figure 9** **Effects of PFI-3 on extracellular $Ca^{2+}$-induced contractions of rat mesenteric arteries.** Original traces showed the effects of PFI-3 on extracellular $Ca^{2+}$-induced contractions in rat mesenteric arteries pretreated by KCl (60 m M) (A) or PE (B) and statistical results. **$P < 0.01$ vs. Ctl ($n = 5$). The error bars represent the SD. Two-tailed unpaired $t$ tests were performed to evaluate the significance of differences. The number of rats or arteries isolated from different rats is indicated as "$n$".

## PFI-3 supressed L-type VDCC currents in A10 cells

The above data showed PFI-3 did not affect intracellular calcium release, but affected extracellular calcium entry. Thus, the effects of PFI-3 on extracellular calcium influx currents were measured by whole-cell patch clamp techniques in A10 cells (*Guan et al., 2006*). Our research showed that PFI-3 inhibited L-type VDCC current densities in A10 cells (Fig. 12).

## DISCUSSION

There is increasing evidence that PFI-3 is involved in various physiological processes, including protecting the activity of endothelial cells by blocking the function of BRG1 (*Zhang et al., 2020*), reducing the differentiation ability of cultured mouse myoblasts into myotubes (*Sharma et al., 2021*), and increasing sensitivity to DNA damage by targeting SWI/SNF (*Lee et al., 2021*). As a selective, effective, and permeable inhibitor, PFI-3 exhibits almost no cytotoxicity in VSMCs.

Our findings demonstrated that PFI-3 produced concentration-dependent vasorelaxation in both intact and denuded endothelium induced by PE and KPSS. As previous studies showed that PFI-3 can alleviate the expression of IL-6 and CCL2 (*Zhang et al., 2020*) and the down-regulation of thrombomodulin in endothelial cells (*Wu et al., 2022*), we assumed PFI-3-mediated vasorelaxation depends on the vascular endothelium. Unexpectedly, we observed that PFI-3 induced vasorelaxation in mesenteric arterial ring independent of endothelium, and further experiments showed that NO/sGC/cGMP pathway was not involved in PFI-3-induced vasorelaxation. Moreover, the vasorelaxant effects of PFI-3 were not altered by pretreatment with $K^+$ blockers (Gli and TEA), indicating

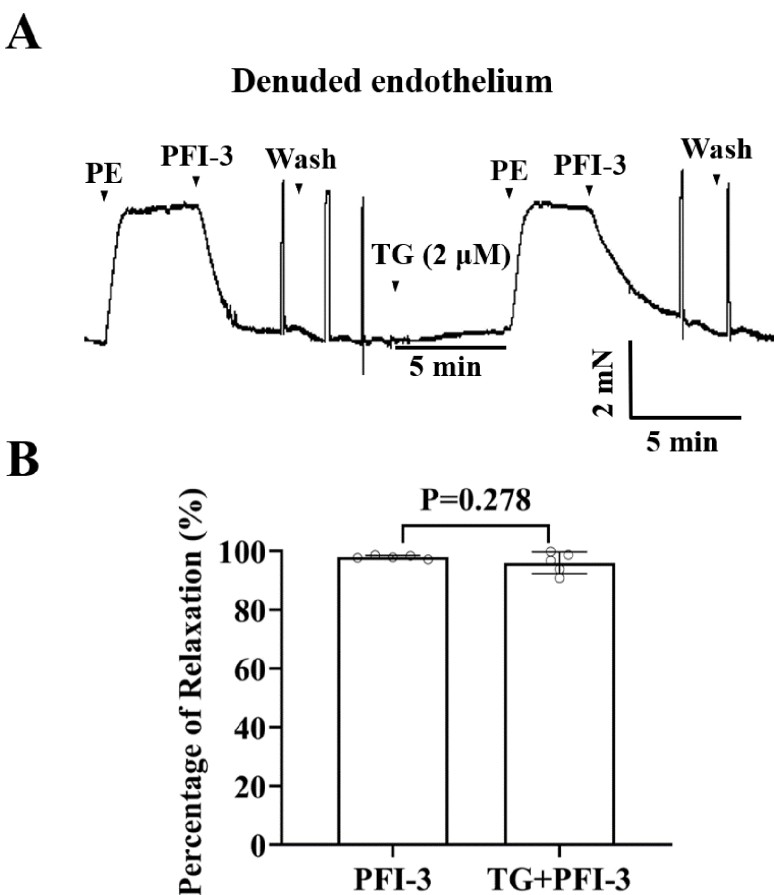

**A**

**Denuded endothelium**

**B**

**Figure 10** **Effects of pretreatment with TG on PFI-3-induced vasodilation.** (A) Original traces and summary data demonstrated that the effect of TG (2 μM) on PFI-3-induced vasorelaxation in rat mesenteric arteries with denuded endothelium following PE-induced constriction. (B) Statistical results. $P = 0.278$ vs. the PFI-3 group ($n = 5$). Error bars are SD. Two-tailed paired $t$ tests were performed to evaluate the statistical difference. $n$, number of rats or arteries isolated from separate rats.

that the vasodilatation induced by PFI-3 was not related to the opening of $Ca^{2+}$-activated $K^+$ (BKCa) or ATP-sensitive potassium ($K_{ATP}$) channels.

Changes in the cytosolic $[Ca^{2+}]_i$ are crucial for regulating VSMC contraction. Extracellular $Ca^{2+}$ influx into VSMCs is mainly mediated by ROCCs and VDCCs, while intracellular $Ca^{2+}$ released from the SR through inositol-1,4,5-trisphosphate receptor and ryanodine receptor channels (*Xu et al., 2012*). PE-induced contraction first involves the release of $Ca^{2+}$ from SR followed by $Ca^{2+}$ influx through ROCCs (*Xia et al., 2008*; *Qin et al., 2014*). In this study, we found that in $Ca^{2+}$-free PSS, PFI-3 abolished vasoconstriction induced by $Ca^{2+}$ of mesenteric arterial rings pre-contracted with PE, which indicated that PFI-3 may affect ROCCs and inhibit $Ca^{2+}$ release from SR stores. Next, our study focused on the impact of TG pretreatment on PFI-3-induced vasorelaxation. TG contributes to $Ca^{2+}$ release through IP3-gated calcium channels, inhibits intracellular $Ca^{2+}$-ATPase, and inhibits $Ca^{2+}$ reuptake into the SR, leading to intracellular $Ca^{2+}$ depletion (*Shen et al., 2013*;

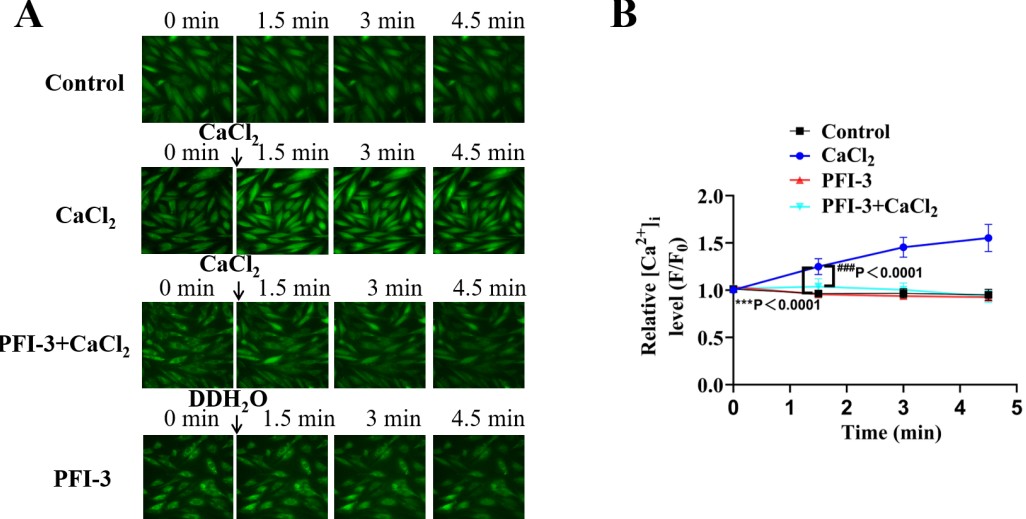

**Figure 11  Effects of PFI-3 on the cytosolic [Ca$^{2+}$]$_i$ in VSMCs.** (A) The cytosolic [Ca$^{2+}$]$_i$ of A10 cells was measured by Fluo-3/AM fluorescent probe and fluorescence microscope. (B) Graph of the data from Fig. 11A. ***$P < 0.0001$ vs. the control group, ###$P < 0.0001$ vs. the CaCl$_2$ group ($n = 30$). The data are represented as the mean ± SD. One-way ANOVA followed by Tukey's *post hoc* test for *post-hoc* comparisons was used. *n*, number of independent cells.

*Li et al., 2017*). However, TG had no significant effect on PFI-3-induced vasorelaxation. These results indicated that PFI-3 can decrease Ca$^{2+}$-induced vasoconstriction *via* ROCCs, but not IP3R channels and the SR Ca$^{2+}$–ATPase pump.

Previous studies reported that high K$^+$-induced arterial ring contraction is mainly due to cell membrane depolarization and extracellular Ca$^{2+}$ influx through L-type VDCCs (*Qin et al., 2014*). Our results showed that PFI-3 inhibited the contraction of mesenteric arterial rings induced by KCl (60 mM) and this effect was accompanied by a decline in extracellular Ca$^{2+}$ influx. Moreover, PFI-3 suppressed L-type VDCC current densities. These findings suggested that PFI-3 may block L-type VDCCs and lead to decrease in extracellular Ca$^{2+}$ influx. That may be a major mechanism involved in the vasorelaxant effects of PFI-3. Although our study demonstrated that the vasodilatory mechanism of PFI-3 is related to the VDCCs and ROCCs in VSMCs, the specific mechanism requires further exploration. By targeting BRDs, PFI-3 specifically inhibits BRG1, which plays a role by affecting gene transcription. Our next study will detect whether the PFI-3 directly or indirectly regulates calcium channels by inhibiting BRG1 to exert its vasodilatory effect. We did not further evaluate the function of PFI-3 induced vasorelaxant *in vivo*, which was a limitation of the present study.

## Statement of ethics

All animal experiments were performed in accordance with the NIH guidelines (Guide for the Care and Use of Laboratory Animals) and approved by the Institutional Animal

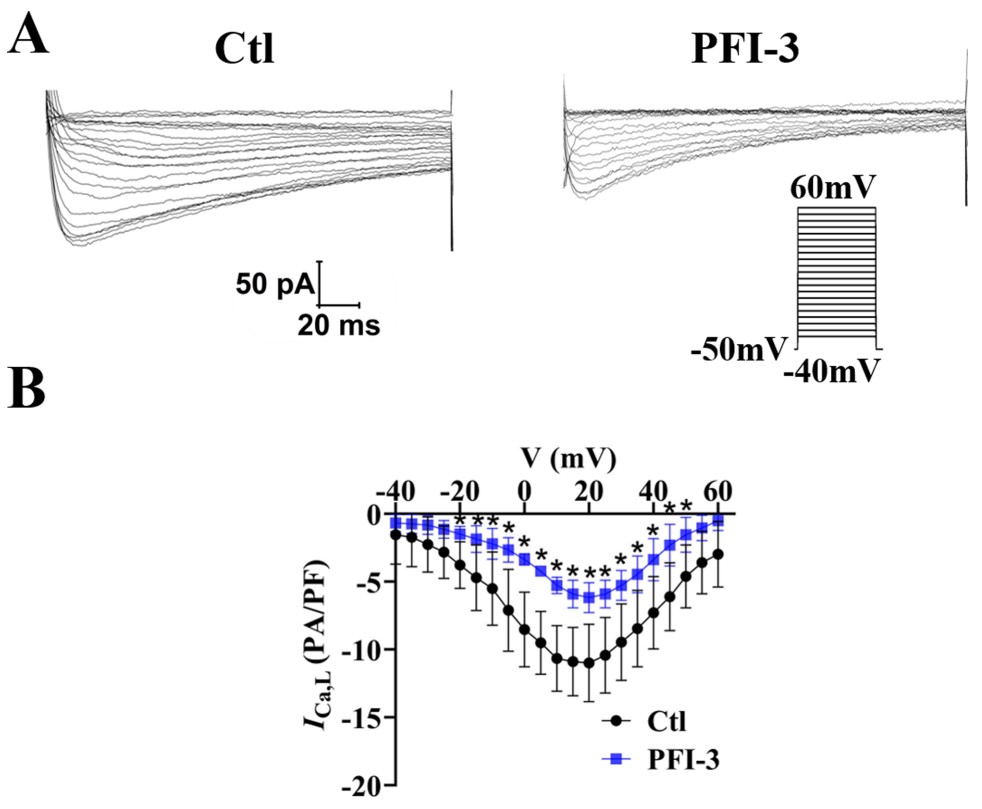

**Figure 12** **Effects of PFI-3 on L-type VDCC activity.** (A) Representative traces of L-type VDCC currents. (B) Current-voltage (I-V) plots for the L-type VDCC currents following treatment with 4 μM of PFI-3 for 20 minutes.*$P < 0.05$ vs. the Ctl group ($n = 5$). Error bars are SD. Two- tailed paired $t$ tests were performed to evaluate the statistical difference. $n$, number of independent cells.

Care and Use Committee of Harbin Medical University (IRB3102619). The adult Sprague-Dawley rats used in this study were provided by the Experimental Animal Center of Harbin Medical University (Grade II).

### Abbreviations

| | |
|---|---|
| **Ach** | Acetylcholine chloride |
| **BRDs** | bromodomains |
| **BRG1** | Brahma-related gene 1 |
| **Gli** | glibenclamide |
| **KPSS** | high-K$^+$ salt solution |
| **PE** | phenylephrine |
| **PSS** | physiological salt solution |
| **ROCCs** | receptor-operated calcium channels |
| **TEA** | Tetraethylammonium |
| **TG** | thapsigargin |
| **VDCCs** | voltage-dependent calcium channels |
| **VSMCs** | vascular smooth muscle cells |

### Funding

This work was supported by the National Natural Science Foundation of China (81872870, 82070312 and 31400983), the Scientific Fund of Heilongjiang Province (H2018011), the Scientific Fund of Heilongjiang Province (LH2022H003), and the Heilongjiang Province Postdoctoral Foundation (LBH-Q19155). The funders had no role in study design, data collection and analysis, decision to publish, or preparation of the manuscript.

### Grant Disclosures

The following grant information was disclosed by the authors:
National Natural Science Foundation of China: 81872870, 82070312, 31400983.
Scientific Fund of Heilongjiang Province: H2018011.
Scientific Fund of Heilongjiang Province: LH2022H003.
Heilongjiang Province Postdoctoral Foundation: LBH-Q19155.

### Competing Interests

The authors declare there are no competing interests.

### Author Contributions

- Jing Li conceived and designed the experiments, performed the experiments, analyzed the data, prepared figures and/or tables, authored or reviewed drafts of the article, and approved the final draft.
- Xue-Qi Liang conceived and designed the experiments, performed the experiments, authored or reviewed drafts of the article, and approved the final draft.
- Yun-Feng Cui conceived and designed the experiments, authored or reviewed drafts of the article, and approved the final draft.
- Yu-Yang Fu performed the experiments, authored or reviewed drafts of the article, and approved the final draft.
- Zi-Yue Ma analyzed the data, authored or reviewed drafts of the article, and approved the final draft.
- Ying-Tao Cui conceived and designed the experiments, performed the experiments, authored or reviewed drafts of the article, and approved the final draft.
- Xian-Hui Dong performed the experiments, analyzed the data, authored or reviewed drafts of the article, and approved the final draft.
- Hai-Jun Huang analyzed the data, prepared figures and/or tables, and approved the final draft.
- Ting-Ting Tong analyzed the data, prepared figures and/or tables, and approved the final draft.
- Ya-Mei Zhu analyzed the data, prepared figures and/or tables, and approved the final draft.
- Ya-Dong Xue analyzed the data, prepared figures and/or tables, authored or reviewed drafts of the article, and approved the final draft.

- Yong-Zhen Wang performed the experiments, prepared figures and/or tables, authored or reviewed drafts of the article, and approved the final draft.
- Tao Ban conceived and designed the experiments, prepared figures and/or tables, and approved the final draft.
- Rong Huo conceived and designed the experiments, authored or reviewed drafts of the article, and approved the final draft.

## Animal Ethics

The following information was supplied relating to ethical approvals (i.e., approving body and any reference numbers):

Institutional Animal Care and Use Committee of Harbin Medical University provided full approval for this research(IRB3102619).

## Data Availability

The raw measurements are available in the Supplementary Tables.

## Supplemental Information

Supplemental information for this article can be found online at http://dx.doi.org/10.7717/peerj.15407#supplemental-information.

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
