# Peer review of "PFI-3 induces vasorelaxation with potency to reduce extracellular calcium influx in rat mesenteric artery"

_PeerJ, doi:10.7717/peerj.15407_

## Round 0.1 · original submission · Major Revisions

When revising your manuscript, please consider all issues mentioned in the comments from the three reviewers carefully and provide suitable responses for any comments. Please note that your revised submission may need to be re-reviewed.

PeeJ values your contribution and I look forward to receiving your revised manuscript.

Reviewer 1 ·

Basic reporting

The manuscript contains many basic language and scientific mistakes, making the paper read like a first stage draft, not a finished article. It is not the reviewer’s role to proofread. Therefore comprehensive lists of problems are not supplied here, and only a few sample problems will be identified. For example, abbreviations were repeated multiple times (eg., PSS in line 102 and 108, VSMC in line 59 and line 200) or words already abbreviated are spelled in full again later (line 112 Sprague-Dawley (SD) rats – abbreviated and line 118 Adult male Sprague-Dawley rats – spelled in full). Also, a space is missing between values and units in several places (eg., 40mg/kg in line 119).
Writing style is verbose and does not comply with the international convention for scientific report based on IMRaD format. In particular, Results are difficult to read because substantial part of writing seems unnecessary, and even if authors consider such information is required, the place to write what seems background is elsewhere. For example, line 156 says
“PE induced vasoconstriction by stimulating α1-adrenergic receptors on the smooth muscle cell membrane”.
This is unnecessary to the readership of PeerJ in Results. A brief description (perhaps in Introduction or Methods) “an α1-adrenergic agonist, PE” should suffice. About a third of current contents could be consolidated and removed. For example, are lines 130-134 repeating experimental procedure narrated in the earlier places necessary?
Perhaps most seriously, the writing lacks accuracy. Although experienced readers would be able to “understand” what authors mean, writing should be as precise as possible to prevent serious misunderstanding. To list just a few examples, line 62 says
“Contraction is stopped through the removal of cytosolic Ca2+ by Ca2+ channel blockers” Ca2+ channel blockers inhibit Ca2+ influx while removal of cytosolic Ca2+ is done by Ca2+ pumps in the SR and cell membrane. Likewise, line 75 says
“leading to the activation of inositol triphosphate on the endoplasmic reticulum (1,4,5-triphosphate inositol, IP3) receptors,” This sentence means that inositol triphosphate, a second messenger found on the ER, is activated. Furthermore, the location of (1,4,5-triphoshate inositol, IP3) in the sentence seems odd. Similarly, line 201 says
“As a results of membrane hyperpolarization, the activation of BKCa channels causes K+ efflux”
The cause and effect are the wrong-way-round here: opening of BCa channels produce K+ efflux, leading to smooth muscle cell membrane hyperpolarization.
Authors could consider asking a scientist who is a native speaker of English to make required modification of the language.

Experimental design

Most of Results were obtained using isometric tension measurement combined with pharmacological tools. Although this type of experiment could potentially offer insight to vascular contraction/relaxation mechanisms, conclusions at cellular and molecular level are rarely gained unless pharmacological tools are extremely specific or more molecular approach (eg., arteries from transgenic animals) is taken. This is perhaps why the A10 cell calcium measurements were performed, but experiments using A10 cells do not solve fundamental issues of lack of mechanistic answers to the hypothesis. For example, Figures 7 and 8 address possible PFI-3 effects on BKCa channels, KATP channels and voltage dependent Ca channels. Why more cellular techniques such as patch clamp that will produce clear results were not used? The volume of data presented notwithstanding; the contents seem rather basic.
A-first-draft problems are also found in the experimental designs. Line 112 says “Adult male Sprague-Dawley (SD) rats (SHR, 12 weeks old”. In terms of laboratory animals, SHR normally stands for “spontaneously hypertensive rat”, a high blood pressure animal model and therefore data obtained from such animals would be non-standard. Is this a typo? Furthermore, contradictory statements are found in the manuscript. For example, line 207 says “after pre-incubating TEA (10mM (sic)), a non-selective calcium-activated K+ channel blocker” while line 265 says “TEA, a BKCa blocker (10 mM)”. Did authors not notice this internal contradiction?

Validity of the findings

Due to the issues described in Experimental Design, contraction experiments require complementing more cellular approaches to validate the findings.

·

Basic reporting

In“The effect and mechanism of PFI-3 on vascular tension”, Jing Li, Xue-Qi Liang et al. found that PFI-3 relaxes rat mesenteric arteries by inhibiting intracellular calcium release and extracellular calcium influx. In terms of mechanism, the vasodilation is likely related to the voltage-dependent calcium channels (VDCCs) and the receptor-operated calcium channels (ROCCs) on vascular smooth muscle. However,PFI-3, as a recently developed monomer compound, its advantages over other compounds have not been mentioned by the authors. In addition, it is unclear whether it will cause additional AEs in vivo.

Experimental design

1. The authors mentioned in the background that PFI-3 has been described as a potential therapeutic drug for vascular endothelial cells, but the experimental results show that PFI-3 induced vasorelaxation is not related to endothelium. For this point, you need to explain your experimental results in more detail.
2. In page 13, line 157, please explain the rationality of the inconsistency between the concentration of PE used to pre-constricted mesenteric artery and the concentration of Ach used to dilate blood vessels.
3. From Figure 1 to Figure 3, why is the concentration of PFI-3 used to prevent vasoconstriction different from that used to relax PE induced rat mesenteric artery contraction.
4. In page 15, line 206, although both BKCa channel and KATP channel are important channels for regulating vascular tension, there is no sufficient reason to speculate that PFI-3 induced vasorelaxation is caused by activation of KATP channel and BKCa.
5. In page 17, line 245, the authors declared that there was no significant change in [Ca2+]c in the group that was only preincubated with PFI-3 (4µM). Please explain whether this proves that PFI-3 (4µM) does not affect the release of intracellular Ca2+.

Validity of the findings

The effect of PFI-3 on vascular tension only through the microvascular tension detection system(DMT) to mesenteric artery vascular tension, and the mechanism of PFI-3 on vascular tension only through the calcium channel and potassium channels. The experimental results are not rigorous enough. It is suggested to increase animal experiments in vivo, further verify the results. Please show that how the 96 rats were grouped in this experiment.

Reviewer 3 ·

Basic reporting

In this study authors reported that PFI-3 relaxes rat mesenteric arteries by inhibiting intracellular calcium release and extracellular calcium influx and its mechanism was related to the voltage-dependent calcium channels (VDCCs) and the receptor-operated calcium channels (ROCCs) on vascular smooth muscle. This is not an interesting and innovation study.
There are problems with the English writing and the quality of the figures.

Experimental design

1. The description of the experimental method is not clear enough.
2. How to determine the working concentration of PFI-3?
3. Authors should double-check the grouping and dosing sequence in Figure 6, 5, 8 and 9. Both preventive and therapeutic approaches should be used to determine the effect of PFI-3.

Validity of the findings

The current data cannot support the conclusions obtained by the authors in the manuscript, and more experiments should be added and more data should be provided in mechanism research.

Additional comments

In this study authors reported that PFI-3 relaxes rat mesenteric arteries by inhibiting intracellular calcium release and extracellular calcium influx and its mechanism was related to the voltage-dependent calcium channels (VDCCs) and the receptor-operated calcium channels (ROCCs) on vascular smooth muscle. This is not an interesting and innovation study. In addition, the study has the following problems.
1. The progress of research on PFI-3 in vascular-related diseases was not adequately described in background, which made the basis of the study inadequate.
2. The description of the experimental method is not clear enough.
3. How to determine the working concentration of PFI-3?
4. Authors should double-check the grouping and dosing sequence in Figure 6, 5, 8 and 9. Both preventive and therapeutic approaches should be used to determine the effect of PFI-3.
6. The current data cannot support the conclusions obtained by the authors in the manuscript, and more experiments should be added and more data should be provided in mechanism research.
7. There are problems with the English writing and the quality of the figures.

---

## Round 0.2 · accepted · Accept

The current version is much better than the original version, and I think it is ready for publication. However, the authors need to proofread the article carefully since some things need to be revised. For example, references 13 and 20 use the full journal name while others do not.

·

Basic reporting

no comment

Experimental design

no comment

Validity of the findings

no comment

Reviewer 3 ·

Basic reporting

no comment

Experimental design

no comment

Validity of the findings

no comment

Additional comments

The authors have responded to the questions raised and revised the manuscript.